# Radical Resection of Malignant Tumors of Major Salivary Glands: Is This Possible?

**DOI:** 10.3390/cancers16040687

**Published:** 2024-02-06

**Authors:** Giulio Cantù

**Affiliations:** Head and Neck Surgery Department, Fondazione IRCCS Istituto Nazionale dei Tumori, 20133 Milan, Italy; gcantu43@gmail.com

**Keywords:** salivary gland tumors, facial nerve, lingual nerve, margins

## Abstract

**Simple Summary:**

Resection of a malignant tumor of the head and neck should be radical, i.e., the specimen should be surrounded by healthy tissue at all its margins. The thickness of this layer of healthy tissue has varied over time. From the initial categorical minimum limit of 5 mm, we have moved on to lower thicknesses, but never less than 1 mm. Can these rules be respected in surgery of the salivary glands, especially of the parotid gland? Owing to the complex branching and connections of the facial nerve within the parotid gland, even a medium-sized malignant tumor may be in contact with a branch of the nerve, thus raising the question of its preservation. The same issue concerns the lingual nerve in the resection of a malignant tumor of the submandibular and sublingual glands. In this article, examining the studies published on this topic, the pros and cons of nerve preservation will be analyzed.

**Abstract:**

In primary therapy, a universally recognized surgical indication applies to all tumors of the salivary glands. According to the classic rule, radical resection of a head and neck tumor requires clean margins of at least 5 mm, although recent studies have shown that for certain locations, 1 mm may be sufficient. In the surgical resection of a tumor of the salivary glands, especially of the parotid gland, can these rules be respected? Owing to the complex branching and connections of the facial nerve within the parotid gland, even a medium-sized malignant tumor may be in contact with a branch of the nerve, thus raising the question of its preservation. The facial nerve is so important from a functional and aesthetic point of view that it is commonly believed that it should be preserved unless it is incorporated into the tumor. This is a compromise between an oncological resection, that is, the complete excision of the tumor with no residual cancer cells left behind, and quality of life. Almost all authors try to overcome this lack of radicality by indicating postoperative (chemo)radiotherapy. In this article, the pros and cons of nerve preservation will be analyzed by examining the published studies on this topic.

## 1. Introduction

Salivary malignant tumors are rare neoplasms accounting for approximately 3% of all head and neck tumors. “Between 64 and 80% of all primary epithelial salivary gland tumors occur in the parotid gland, with most located in the superficial (lateral) lobe, 7–11% occurring in the submandibular glands, fewer than 1% occurring in the sublingual glands, and 9–23% occurring in the minor glands” [1]. Owing to this rarity, the understanding of this disease has been based mostly on clinical series rather than on randomized evidence, which is unlikely to emerge for these tumors [2]. Most tumors are benign (54–79%) and there is a great variation in the malignant/benign ratio between the various sites (15–32% in the parotid, 41–45% in the submandibular, 70–90% in the sublingual and 50% in the minor salivary glands) [1]. Despite this, “perhaps no tissue in the body is capable of producing such a diverse histopathological expression than salivary tissue” [3]. In his text, Batsakis described 18 types of benign and 21 malignant tumors [3]. The most recent WHO classification of 2017 reports 23 malignant and 11 benign histological types [4].

Regardless of the histological type and site of the tumor, the primary therapeutic indication is surgical resection (when possible). The National Comprehensive Cancer Network (NCCN) Guidelines 2019 include the following statement: “The major therapeutic approach for salivary gland tumors is adequate and appropriate surgical resection. Surgical intervention requires careful planning and execution, particularly in parotid tumor surgery because the facial nerve is in the gland” [5]. Certainly, the two adjectives “adequate” and “appropriate” were carefully chosen because radical resection of a parotid tumor according to the most accredited rule of radicality (free margin ≥ 5 mm) is often impossible without sacrificing some branch of the facial nerve.

This narrative review intends to analyze the pros and cons of preserving the facial and lingual nerves in the surgical removal of malignant tumors of the major salivary glands.

## 2. Radical Resection of a Malignant Parotid Tumor: A Fact or a Myth?

There have been two dogmas in oncological surgery in the past: “block resection” and “radical resection”. In the 1970s, when I was a trainee at the National Cancer Institute of Milan, Italy, the two aforementioned concepts were summarized by my teachers in an aphorism: “A tumor of the head and neck must be removed without seeing it, as it must be surrounded by healthy tissue in all its boundaries”.

The concept of block resection has been overcome. A long series of studies has demonstrated that carcinomas of some sites of the head and neck can be treated with piecemeal removal, provided that healthy tissue is reached at all the margins of the resection.

Wolfgang Steiner was the pioneer in applying piecemeal resection in the treatment of laryngeal cancer, initially for early cases and later also in advanced stages. Between 1979 and 1991, he treated 240 patients with laryngeal cancer in this way, and in 1993, he published the results [6]. There were only six local recurrences (2.5%), with one patient needing total laryngectomy. The adjusted five-year survival rate was 100%. The path of progress of this new approach has not been without contention and has remained controversial in some quarters. In fact, the technique was strongly resisted, even derided at the beginning [7]. The results of subsequently published studies have further demonstrated the validity of the technique, and today, no one questions it.

The piecemeal resection technique was subsequently also applied in the removal of other head and neck tumors, such as those of the ethmoid extending to the anterior skull base [8].

However, if the dogma of block resection has collapsed, radical resection of the tumor with margins of healthy tissue remains a dogma.

The most important question in this regard is as follows: When is a surgical resection radical?

The definition of radical resection and the thickness of healthy tissue around the tumor are highly controversial topics. The 2005 guidelines of the Royal Society of Pathology defined a pathological surgical margin of more than 5 mm as clear, a margin of 1–5 mm as close, and a margin less than 1 mm as positive [9]. Additionally, the 2014 NCCN guidelines defined a clear surgical margin as histological confirmation of a distance of at least 5 mm from the invasive tumor to the resected margin [10]. This statement is still considered valid by many oncologists and is sometimes the source of heated discussion during multidisciplinary head and neck cancer tumor board meetings when deciding on possible postsurgical (chemo)radiotherapy. In fact, in addition to the well-known shrinkage of the margins of specimens for formalin fixation and slide preparation, some studies involving a large number of patients have demonstrated that these subdivisions are too categorical, especially for oral cancers [11,12]. Zanoni et al. analyzed 381 patients who underwent surgical resection for squamous cell carcinoma of the oral tongue. They found that local recurrence-free survival was significantly affected only by surgical margins less than or equal to 2.2 mm. The authors concluded that “This new definition of close margins stratifies the risk for local recurrence better than the arbitrary 5.0-mm cutoff that has been used” [11]. Another study with 432 patients showed that the cutoff can even be set at 1 mm [12]. Even for laryngeal carcinomas resected via the transoral endoscopic technique, the margins are generally defined as positive if they are infiltrated by neoplastic tissue and close if <1 mm of free tissue is present between the margins and the tumor [13].

Is it possible to apply the aforementioned parameters to salivary gland tumors and, in particular, those of the parotid gland? The answer is that this is often not possible. Owing to the complex branching and connections of the facial nerve within the parotid gland, even a medium-sized benign tumor may be in contact with a branch of the nerve. This possibility almost always becomes common in the case of malignant tumors. Therefore, according to the above rules for radical resection, even for less stringent ones, one or more branches of the nerve should be sacrificed.

However, the facial nerve is so important from an aesthetic and functional point of view for quality of life that almost all authors agree not to damage it. In the NCCN guidelines [5], there are two rather ambiguous statements: “The nerve should be preserved if it is not directly involved by the tumor”; “The facial nerve should be sacrificed if there is preoperative nerve involvement with palsy or if there is direct invasion of the tumor into the nerve where the tumor cannot be separated from the nerve”.

However, what does “directly involved” mean? When the nerve, although functioning well, is tenaciously adherent to the tumor and its peeling is difficult, is it or is it not “directly involved”? Behavior, such as that indicated in the guidelines, is a compromise between the dogma of clean margins and aesthetic and functional results.

To overcome this uncertainty of having performed a radical resection, almost all authors indicate postoperative radiotherapy treatment. It is difficult to say what impact this additional treatment has on patients’ quality of life. Many studies have reported negative consequences on the quality of life linked to parotid irradiation in head and neck carcinomas in general and, above all, in nasopharyngeal carcinoma [14,15]. Regarding malignant tumors of the parotid gland, almost all studies showed that postoperative radiotherapy improves local control, especially for high-grade and advanced T3 and T4 carcinomas. Zbaren et al. even suggested this treatment for all T2 carcinomas and high-grade T1 carcinomas [16].

On the other hand, few studies report the sequelae of postoperative radiotherapy after parotidectomy on quality of life, which, however, appear to be modest. Al-Mamgani et al. wrote, “The 5-year cumulative incidence of grade ≥2 late toxicity was 8%. QOL scores deteriorate during and shortly after treatment but returned in almost all scales to baseline scores within 6 months” [17].

Many published studies on parotidectomy circumvent the problem of clean margins by not reporting margin status. The few studies that have examined these data incredibly but logically report high rates of lack of radicality. Garden et al. [18] reported that 83/198 (42%) patients with adenoid cystic carcinoma (ACC) of major and minor salivary glands had microscopic positive margins, and an additional 55 (28%) had close or uncertain margins. Erovic et al. evaluated 215 patients with malignancies of the parotid gland and found positive margins (i.e., tumors extending to the inked margin of the specimen) in 38.9% of patients who underwent parotidectomy. Positive margins were noted in 34.8% of patients with grade I tumors, 32.3% of patients with grade II tumors, and 45.1% of patients with grade III tumors [19]. Morse et al. examined 5639 cases of parotid malignancies in the National Cancer Database (NCDB) and presented the results of what is, to my knowledge, the largest multi-institutional series. The overall percentage of positive margins was 31%. They also found that ACC was associated with an increased rate of positive margins. Another important finding was that treatment at academic or high-volume facilities was associated with a decreased rate of positive margins [20]. This finding demonstrates once again that removing a malignant tumor of the parotid gland is not an easy surgical procedure [20].

## 3. Submandibular and Sublingual Glands

Although, as previously mentioned, the percentage of tumors in the submandibular gland is lower than that in the parotid gland and in the sublingual gland is negligible (approximately 1% of salivary tumors), the percentage of malignant tumors is greater in these locations than in the parotid gland (20% versus 50 and 80%, respectively [5]. In particular, ACC is uncommon in the parotid gland. Its incidence in the large historical series presented by Eneroth [21] (one of the historic fathers of salivary gland surgery) and by Foote and Frazell [22] (802 and 776 cases, respectively) lies between 6% and 15% of all malignancies. On the contrary, ACC is much more frequent in the submandibular gland. In two series studied by Spiro et al. [23] and by Eneroth et al. [24], the incidence of ACC was 35% and 40%, respectively. A recent multicenter study instead reported a higher percentage of ACC in the parotid gland than in the submandibular gland: 19.5% and 11.5%, respectively [25].

Radical resection of the submandibular salivary gland for a malignant tumor is apparently less problematic. However, if this is true, why do all the studies on malignant tumors of the salivary glands report a higher rate of local recurrence and lower survival for patients with malignancies of the submandibular gland than for those with malignancies of the parotid gland [3,23]? For example, in Aro’s series, there were 44% relapses of which 16% were local [26]. Batsakis [3] reported data from Eneroth, who analyzed hundreds of malignant tumors of the salivary glands, and reported 5- and 10-year survival rates for ACC of the parotid gland of 73% and 39%, respectively, while the corresponding rates for the submandibular gland were 50% and 25%, respectively. The 15-year survival rate was 21% versus 0%. These data were confirmed by a series of subsequent studies [27,28].

The incidence of neck metastases in ACC of the head and neck is low, and the reported incidence ranges from 0 to 18%. However, when the primary tumor is in the submandibular gland, the incidence is approximately 34% [29]. Among 483 patients with major salivary gland ACC, the International Head and Neck Scientific Group reported that the prevalence of positive nodes was 14.5% for the parotid gland, 22.5% for the submandibular gland, and 24.7% for the sublingual gland [30]. Fang et al. analyzed 914 patients with submandibular gland carcinoma selected from the Surveillance, Epidemiology, and End Results (SEER) database. Lymph node (LN) metastasis developed in 382 (41.8%) patients. The number of metastatic LNs was one in one hundred and three (11.3%) patients, two in seventy-five (8.2%), three in forty (4.4%), four in nineteen (2.1%), five in twenty-three (2.5%), six in twenty-three (2.5%), and seven or more in ninety-nine (10.8%) [31].

This high rate of lymph node metastasis may explain the regional recurrence, the distant metastasis, and the lower survival of patients with submandibular ACC but does not explain the high rate of local recurrence. While hundreds of studies have been performed on the behavior of the facial nerve during parotidectomy, little has been written about the role of the lingual nerve in the removal of tumors of the submandibular and sublingual glands. Many studies have reported nerve involvement by ACC as a negative prognostic factor, particularly if the nerve involved is above a “certain size” or “named nerve” [18,32]. From this point of view, the most studied nerve is the trigeminal nerve, especially its third branch. Parker et al. [33] wrote that both anterograde and retrograde perineural tumor spread were seen, although retrograde spread was significantly more common. In fact, many cases of ACC of the minor salivary glands of the oral cavity and paranasal sinuses with extension to the cavernous sinus and Gasser’s ganglion have been described [34].

Is the lingual nerve, which originates from the third branch of the trigeminal nerve and is above a “certain size”, a “name nerve” or not?

In the surgical resection of benign tumors and in nonneoplastic pathology of the submandibular gland, the lingual nerve is visualized and respected, the mylohyoid muscle is retracted, and the anterior part of the gland surrounding Wharton’s duct is left in place. Is this oncologically correct in malignant tumors? In a nice narrative review on sublingual gland malignancies, Park et al. performed a PubMed search, analyzed 14 articles, and found that malignant tumors of the sublingual gland often extend to the submandibular gland (and vice versa), thus incorporating the lingual nerve; therefore, most surgeons advocate concurrent resection of the lingual nerve during curative surgery because the rates of invasion of ACC and ACC more than 4 cm in the largest dimension were 30.8% and 42.9%, respectively [35]. Therefore, does not removing the lingual nerve allow for oncological radicality? A fact that should give pause for thought is that reported by Benchetrit et al. These authors analyzed 1150 patients in the NCDB from 2004 to 2014 who were diagnosed with submandibular carcinoma and underwent primary surgical resection. The percentage of patients with a positive margin was 41.0%. Increased odds of positive margins were observed in patients with advanced T-stage disease and ACC histology. Positive margins were associated with reduced overall survival (OS) (58% vs. 69% 5-year OS, *p* < 0.001) when controlling for patient, tumor, and management factors [36]. Perhaps we should ask ourselves how many of these involved margins are due to the non-removal of structures surrounding the tumor that may be infiltrated (the lingual nerve, mylohyoid muscle, etc.).

I searched in vain for studies on the consequences of resection of the lingual nerve in patients with carcinomas of the submandibular gland. I found many studies on the effect of resection of this nerve, with or without simultaneous resection of the hypoglossal nerve, for large tumors of the oropharynx. Elfring et al. wrote that “Transected lingual and hypoglossal nerves were associated with difficulty in swallowing, social eating, dry mouth and social contact” [37]. The same team, in a previous article, wrote that “During base of tongue reconstruction the lingual nerve is often severed on one or both sides, affecting sensation in the preserved tissue of the anterior tongue. The loss of specific tongue sensations could negatively affect a person’s oral function and quality of life” [38].

It is clear that any surgical resection for malignant tumors, even if the best reconstruction is carried out, involves an alteration of some physiological function and a more or less serious worsening of the patient’s quality of life. The surgeon must therefore always evaluate the pros and cons of each operation.

## 4. Conclusions

In conclusion, surgeons should carefully evaluate the pros and cons of sacrificing the aforementioned nerves. Certainly, the facial nerve is so important for quality of life that any attempt to preserve it can be justified. Does the same attitude also apply to the lingual nerve?

It would be appropriate to clarify these issues to the patient in a simple and understandable way. Furthermore, all studies of malignant tumors of the salivary glands should unambiguously report the status of the resection margins.

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
