# Peer review of "Radical Resection of Malignant Tumors of Major Salivary Glands: Is This Possible?"

_cancers, 2024, doi:10.3390/cancers16040687_

Round 1

Reviewer 1 Report

Comments and Suggestions for Authors

The author provides an interesting discussion on the pros and cons of nerve preservation in the surgery of salivary gland tumors. However, the review would be significantly improved by also discussing the pros and cons of post-operative radiotherapy in relation to conservative surgery. What is known for this therapeutic regimen with respect to patient survival and quality of life/long-term negative effects of those afflicted by malignant salivary gland tumors?

Row 10-11, The sentence should be rephrased to more clearly state a question. For example: “Can these rules be respected in surgery of salivary glands, especially of the parotid gland?” 

Row 43, To maintain objectivity, the word “wonderful” should be removed.

Row 76, The word “anymore” is superfluous and should be removed.

Row 97, Should state: “2.2 mm”

Row 140-145, A recent multi-institutional study by Persson et al. (J Pathol 2023 PMID: 37565350 ) included the localization of 391 primary ACCs.  The authors reported that parotid ACCs were present in up to 20% of the cases and that submandibular ACC was rarer with an incidence of 12% in the studied cohort. The results from this study should be cited in the discussion.

Rows 261-290, The reference numbering after reference 20 is incorrect.

Author Response

Reviewer 1

Dear reviewer, thank you for your careful consideration of my manuscript. I have carefully read your comments aimed at improving its quality. Below you can find my point-by-point answers and the text with the additions/corrections.

Best regards.

Giulio Cantù

Comments and Suggestions for Authors

The author provides an interesting discussion on the pros and cons of nerve preservation in the surgery of salivary gland tumors. However, the review would be significantly improved by also discussing the pros and cons of post-operative radiotherapy in relation to conservative surgery. What is known for this therapeutic regimen with respect to patient survival and quality of life/long-term negative effects of those afflicted by malignant salivary gland tumors?

I added a paragraph on postoperative radiotherapy and its late effects with 4 additional references

Row 10-11, The sentence should be rephrased to more clearly state a question. For example: “Can these rules be respected in surgery of salivary glands, especially of the parotid gland?” I corrected the sentence according to your suggestion

Row 43, To maintain objectivity, the word “wonderful” should be removed. “wonderfull” has been removed.

Row 76, The word “anymore” is superfluous and should be removed. “Anymore” has been removed.

Row 97, Should state: “2.2 mm” I corrected cm to mm

Row 140-145, A recent multi-institutional study by Persson et al. (J Pathol 2023 PMID: 37565350 ) included the localization of 391 primary ACCs.  The authors reported that parotid ACCs were present in up to 20% of the cases and that submandibular ACC was rarer with an incidence of 12% in the studied cohort. The results from this study should be cited in the discussion.

I added the sentence: " A recent multicenter study instead reported a higher percentage of ACC in the parotid gland than in the submandibular gland, 19.5% versus 11.5%, respectively”. I also added the relevant reference.

Rows 261-290, The reference numbering after reference 20 is incorrect. The reference numbers have been corrected, even for new entries

Reviewer 2 Report

Comments and Suggestions for Authors

The article explores the challenges and considerations in the surgical resection of malignant salivary tumors in the head and neck, particularly focusing on the parotid gland. The primary emphasis is on the dilemma of achieving radical resection with clean margins while preserving crucial nerves like the facial nerve. Additionally, it addresses the less-explored aspect of lingual nerve preservation during the resection of malignant tumors in the submandibular and sublingual glands.

Here are some potential comments for enhancing the article:

-The use of terms such as "radical resection," "clean margins," and "healthy tissue" could benefit from clear definitions or explanations for readers.

-The article might benefit from a more distinct and organized structure. Consider breaking down the content into subsections to make it easier for readers to follow the logical flow of information.

-Consider incorporating figures or graphs to illustrate the complex branching of the facial nerve and its anastomoses with other cranial nerves. This addition could enhance the article's accessibility and engagement.

-The section discussing the lingual nerve could be further developed. Provide more context on its role, potential complications during surgical resection, and any existing literature on the preservation or sacrifice of the lingual nerve in similar cases.

-Ensure consistent use of terminology throughout the article. For instance, terms like "radical resection" and "clean margins" should be consistently defined and used to avoid confusion.

-Consider a thorough proofreading by a native speaker to enhance the article's flow. There are instances where the text may appear redundant.

Author Response

Reviewer 2

Dear reviewer, thank you for your careful consideration of my manuscript. I have carefully read your comments aimed at improving its quality. Below you can find my point-by-point answers and the text with the additions/corrections.

Best regards.

Giulio Cantù

Comments and Suggestions for Authors

The article explores the challenges and considerations in the surgical resection of malignant salivary tumors in the head and neck, particularly focusing on the parotid gland. The primary emphasis is on the dilemma of achieving radical resection with clean margins while preserving crucial nerves like the facial nerve. Additionally, it addresses the less-explored aspect of lingual nerve preservation during the resection of malignant tumors in the submandibular and sublingual glands.

Here are some potential comments for enhancing the article:

-The use of terms such as "radical resection," "clean margins," and "healthy tissue" could benefit from clear definitions or explanations for readers.

Although these terms are commonly used in oncological surgery, I have added a definition of what is meant by oncologically radical resection

-The article might benefit from a more distinct and organized structure. Consider breaking down the content into subsections to make it easier for readers to follow the logical flow of information.

I added the subsection "Submandibular and sublingual glands"

-Consider incorporating figures or graphs to illustrate the complex branching of the facial nerve and its anastomoses with other cranial nerves. This addition could enhance the article's accessibility and engagement.

I agree that some pictures would enhance the article aesthetically. However, I have assumed that all head and neck surgeons are familiar with the anatomy of the facial and lingual nerves. Furthermore, since I am not an anatomical designer, this would have meant asking permission to reproduce figures from other texts.

-The section discussing the lingual nerve could be further developed. Provide more context on its role, potential complications during surgical resection, and any existing literature on the preservation or sacrifice of the lingual nerve in similar cases. I added a paragraph on this issue

-Ensure consistent use of terminology throughout the article. For instance, terms like "radical resection" and "clean margins" should be consistently defined and used to avoid confusion. See above

-Consider a thorough proofreading by a native speaker to enhance the article's flow. There are instances where the text may appear redundant.

The manuscript has been reviewed by AJE, as per the attached certificate.

Round 2

Reviewer 1 Report

Comments and Suggestions for Authors

I am happy with the revisions made by the author.